# Semi-Supervised Generative Models for Multi-Agent Trajectories

**Dennis Fassmeyer**
Leuphana University of Lüneburg
`dennis.fassmeyer@leuphana.de`

**Pascal Fassmeyer**
Leuphana University of Lüneburg
`pascal.fassmeyer@leuphana.de`

**Ulf Brefeld**
Luephana University of Lüneburg
`brefeld@leuphana.de`

## Abstract

Analyzing the spatiotemporal behavior of multiple agents is of great interest to many communities. Existing probabilistic models in this realm are formalized either in an unsupervised framework, where the latent space is described by discrete or continuous variables, or in a supervised framework, where weakly preserved labels add explicit information to continuous latent representations. To overcome inherent limitations, we propose a novel objective function for processing multi-agent trajectories based on semi-supervised variational autoencoders, where equivariance and interaction of agents are captured via customized graph networks. The resulting architecture disentangles discrete and continuous latent effects and provides a natural solution for injecting expensive domain knowledge into interactive sequential systems. Empirically, our model not only outperforms various state-of-the-art baselines in trajectory forecasting, but also learns to effectively leverage unsupervised multi-agent sequences for classification tasks on interactive real-world sports datasets.

## 1 Introduction

Analyzing the spatiotemporal behavior of multiple agents bears a great deal of value in many domains such as autonomous driving [45, 15], public transportation [29, 54], migration [37, 23] or sports analytics [50, 11]. Particularly, the detection of collective patterns across time and space is of interest to many communities but non-trivial inter-dependencies between agents render estimating the inherent multi-modal distributions often difficult.

Existing generative approaches to modeling sequential multi-agent data thus capitalize on variants from graph networks [52, 14] and (sequential) variational autoencoders [25, 8]; being purely unsupervised, their functionality is grounded in uncovering the implicit modular structure in agent-wise latent variables directly from the observed trajectories via approximate inference [27, 48, 55, 17, 16, 32]. Although a generative view on the problem is very appealing, approaches in this regime either lack interpretable and controllable latent factors (e.g., [55]), thus being limited in practical settings, or put a strong focus on relationship discovery (e.g., [27]), which limits their generative capacity.

An alternative is offered by supervised approaches that aim to counteract these issues by incorporating discrete behavioral indicators into the generation process [60, 61, 49]. Ideally, these labels encode implicit expert knowledge that would be learned by the model and substantially expand the scope of information captured in latent space. Instead of arguing in favor of expensive and tedious manual annotations of spatiotemporal data, however, these approaches propose to incorporate heuristic

36th Conference on Neural Information Processing Systems (NeurIPS 2022).

surrogates as makeshift labels into the problem [62, 60, 38]. While reporting impressive predictive accuracies, they require the existence of label sequences for all training instances and cannot include inexpensive and possibly abundant unlabeled data due to their purely supervised nature.

To close this gap, we propose semi-supervised variational autoencoders for spatiotemporal multi-agent problems to process (weakly) labeled as well as unlabeled data. Our contribution generalizes the class of semi-supervised variational autoencoders [26] to spatiotemporal domains by using ideas from variational recurrent neural networks [8]. At a high level, the derived objective function subsumes previous approaches into a unified framework via a disentangled latent space that can incorporate supervision signals in arbitrary quantities for structuring the discrete latent subspace. Additional adaptive graph network layers provide order in- and equivariance of agents and render the proposed approach appropriate for multi-agent scenarios. The resulting semi-supervised approach is applicable to a wide range of problems including the generation of collective movements as well as classifying situations of interest. Empirically, our contribution significantly advances recent baselines in generation and classification tasks on interactive real-world data.

## 2 Related Work

**Semi-supervised generative models**  Semi-supervised variational autoencoders have been originally studied by [26] who propose to incorporate additional labels in latent space. Related approaches introduce auxiliary variables that leave the original model unchanged but increase the flexibility of the variational posterior [33]. [24] propose to explicitly capture label characteristics in latent space instead of the label values themselves and [39] study generalizations that allow for learning more complex dependency structures. However, all the above approaches are introduced only for static domains. Our contribution constitutes a generalization of the above approaches to spatiotemporal domains.

**Sequential generative models**  Approaches designed for predicting agent trajectories are frequently based on ideas from sequential generative models to encode temporal dependencies and address the stochasticity inherent in future predictions. Sequential extensions of variational autoencoders with dynamic latent variables include [4, 8, 13, 1]. Particularly, the VRNN [8] is related to our contribution, as both deploy a VAE instance per time step conditioned on a recurrent state. However, their work lacks modeling of a social dimension as well as means to integrate (potentially expensive) discrete behavioral indicators. Other approaches associate a single global latent variable with each sequence [5, 18, 10, 9].

**Graph neural networks**  Deep graph-based (GNNs) approaches [46] are a natural methodological choice for applications requiring modeling datasets composed of sets. GNNs operate by learning a chain of hidden representations for each node through an iterative process that relies on aggregating messages along edges in each network layer. Prominent instantiations such as GCN [28], GraphSAGE [19], GAT [52], and others [14, 51, 21, 3, 59, 53] mainly differ in their notion of message passing and neighborhood aggregation [56]. GNN layers can be stacked and allow for context-based and adaptive neighborhoods for each node by incorporating particularly designed skip connections [53]. See [7] for a discussion on the function approximation capabilities of this neural network family.

**Generative models for sequential multi-agent data**  Given the potential benefit across various domains, a great deal of recent publications focus on movements of pedestrians or self-driving cars [35, 45, 15, 36, 57, 44, 2, 22, 58, 31]. [34], however, show that these standard benchmark datasets generally exhibit weak social interaction patterns, so in the remainder, we focus on methods using team sport data. Related approaches show that learning and computing realistic rollouts significantly benefit from the incorporation of makeshift annotations and heuristic surrogates into the problem [62, 60, 38]. [12, 32] propose an architecture based on conditional VAEs [47] to model distributions of future movements of basketball players. [55, 48] combine ideas from graph networks [14] and VRNNs [8] into a unified framework aiming at modeling data from basketball and soccer. Other related approaches [30, 27, 17, 16] aim to explicitly infer discrete latent variables to represent interaction types while executing a trajectory prediction task.

Hence, existing generative models for spatiotemporal pattern detection focus on multi-agent trajectory prediction in fully-(un)supervised settings. Our contribution represents the first semi-supervised generalization of variational autoencoders for spatiotemporal domains, which allows addressing tasks beyond representational or generative modeling involving expensive annotation processes.

## 3 Methods

### 3.1 Preliminaries

Variational autoencoders (VAEs) [25, 40] allow to capture intrinsic conditional dependencies and are often used in structured problems. Essentially, they consist of a $\theta$-parameterized generative model $p_\theta(\boldsymbol{x}, \boldsymbol{z}) = p(\boldsymbol{z})p_\theta(\boldsymbol{x}|\boldsymbol{z})$ and an associated $\phi$-parameterized variational distribution $q_\phi(\boldsymbol{z}|\boldsymbol{x})$. Optimal parameters $\{\theta, \phi\}$ are obtained via maximizing the variational lower bound on the marginal likelihood, given by

$$\log p_\theta(\boldsymbol{x}) \geq \mathbb{E}_{q_\phi(\boldsymbol{z}|\boldsymbol{x})}[\log p_\theta(\boldsymbol{x}|\boldsymbol{z})] - \mathcal{KL}[q_\phi(\boldsymbol{z}|\boldsymbol{x}) \parallel p(\boldsymbol{z})].$$

It is often helpful to augment partially observed discrete labels $y$ into the model to guide the generating process [26]. The resulting model $p(\boldsymbol{x}, \boldsymbol{z}, y) = p(\boldsymbol{x}|\boldsymbol{z}, y)p(y)p(\boldsymbol{z})$ then acts on labeled and unlabeled instances. Consequentially, the variational distribution $q_\phi(y, \boldsymbol{z}|\boldsymbol{x}) = q_\phi(y|\boldsymbol{x})q_\phi(\boldsymbol{z}|y, \boldsymbol{x})$ needs to be defined over both quantities ($\boldsymbol{z}$ and $y$). *Semi-supervised VAEs (SSVAE)* are trained by maximizing

$$\sum_{(\boldsymbol{x}, y)} \left(\mathcal{L}(\boldsymbol{x}, y) + \lambda \log q_\phi(y|\boldsymbol{x})\right) + \sum_{\boldsymbol{x}} \mathcal{U}(\boldsymbol{x}),$$

where $\mathcal{L}(\boldsymbol{x}, y) = \mathbb{E}_{q_\phi(\boldsymbol{z}|\boldsymbol{x}, y)}[\log p_\theta(\boldsymbol{x}|y, \boldsymbol{z}) + \log p(y) + \log p(\boldsymbol{z}) - \log q_\phi(\boldsymbol{z}|\boldsymbol{x}, y)]$ and $\mathcal{U}(\boldsymbol{x}) = \sum_y q_\phi(y|\boldsymbol{x})(\mathcal{L}(\boldsymbol{x}, y)) + \mathcal{H}(q_\phi(y|\boldsymbol{x}))$ denote (variational) lower bounds on labeled and unlabeled instances, respectively, and $\lambda$ is a balancing term. The original motivation for this framework was inspired by semi-supervised classification tasks. However, SSVAEs are also frequently employed for learning meaningful representations and generating new data, thus rendering it a suitable methodological foundation for performing diverse sets of tasks. The dependency structures of $q_\phi$ and $p_\theta$ are displayed visually as part of Figure 1.

### 3.2 Problem Formulation

We are interested in modeling spatiotemporal multi-agent scenarios and focus on positions $\boldsymbol{x}_t^{(a)} \in \mathbb{R}^2$ of an agent $a \in \mathcal{A}$ over time $1 \leq t \leq T$, denoted by $\boldsymbol{x}_{\leq T}^{(a)}$. We assume that a non-empty subset of agents in $\mathcal{A}$ is present in any observed segment and collect their trajectories (in random order) to form collective movements $\boldsymbol{x}_{\leq T} := \{\boldsymbol{x}_{\leq T}^{(a)} \mid a \in \mathcal{A}\}$. We will make use of velocities $d\boldsymbol{x}/dt$ and linearized motion $\Delta\boldsymbol{x}_t = \boldsymbol{x}_{t'} - \boldsymbol{x}_t$, for $t' > t$, of agents in the remainder.

In addition, we introduce discrete labels $y_t^{(a)} \in \mathcal{Y}$ associated with agent $a \in \mathcal{A}$ at time $t$. The collection of sequences $\boldsymbol{y}_{\leq T} := \{\boldsymbol{y}_{\leq T}^{(a)} \mid a \in \mathcal{A}\}$ may be best thought of as discriminative causes of variation in the corresponding trajectories $\boldsymbol{x}_{\leq T}$. In practice, $\boldsymbol{y}_{\leq T}$ may contain anything that aids in generating multi-agent rollouts, including behavioral indicators, long-term goals, or current tasks. Finally, observations $\boldsymbol{x}_{\leq T}$ accompanied by labels $\boldsymbol{y}_{\leq T}$ assemble the labeled share of the data, denoted by $\mathcal{D}_L = \{(\boldsymbol{x}_{\leq T}, \boldsymbol{y}_{\leq T})_n\}_{1 \leq n \leq N}$, while unlabeled observations are collected in $\mathcal{D}_U = \{(\boldsymbol{x}_{\leq T})_m\}_{1 \leq m \leq M}$. We refer to all data by $\mathcal{D} = \mathcal{D}_L \cup \mathcal{D}_U$.

We aim at estimating the underlying distribution that generated the observations in $\mathcal{D}$. A straight forward solution is to maximize the log likelihood given by

$$\log p(\mathcal{D}) = \sum_{\mathcal{D}_L} \log p_\theta(\boldsymbol{x}_{\leq T}, \boldsymbol{y}_{\leq T}) + \sum_{\mathcal{D}_U} \log p_\theta(\boldsymbol{x}_{\leq T}). \tag{1}$$

However, human movement is inherently non-deterministic and multi-modal. Thus, it is common to model the stochasticity inherent in future agent behaviors via latent representations of VAEs. We propose to lower-bound Eq. 1 as follows.

### 3.3 SSVAEs for Modeling Multi-Agent Trajectories

We associate disentangled latent realizations $\{\boldsymbol{z}_t, y_t\}$ with each time step $t$ of the segment while injecting the dependency structures introduced in Section 3.1 for the generative and inference parts. To model temporal dependencies, we additionally condition the model components on the past

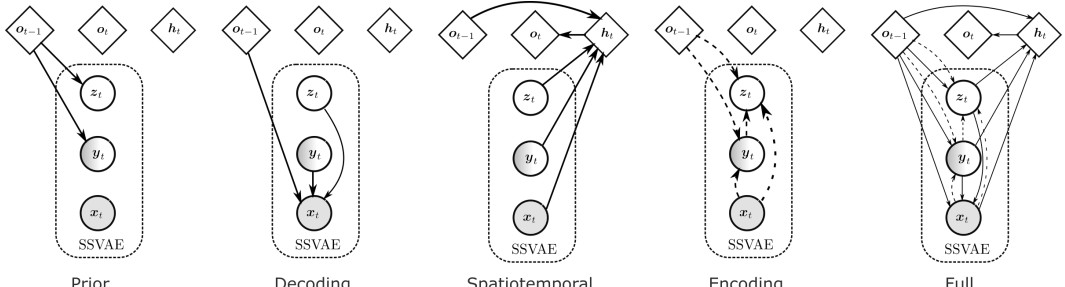

Figure 1: Illustrative graphical model at $t$. Dashed lines indicate the encoding procedure, and solid lines the generative model. We use a semi-supervised VAE (SSVAE, Section 3.1) per time step and additionally introduce variables $\boldsymbol{h}_t$ and $\boldsymbol{o}_t$. The representations in set $\boldsymbol{o}_t$ jointly encapsulate both temporal and spatial patterns. The proposed model can accommodate all contingent proportions of supervision for the discrete latent subspace $\boldsymbol{y}_t$.

observations $\boldsymbol{x}_{<t}$ and latent variables $\boldsymbol{z}_{<t}$ and $\boldsymbol{y}_{<t}$ (cmp. [8]). The joint distribution factorizes into

$$p_\theta(\boldsymbol{x}_{\leq T}, \boldsymbol{z}_{\leq T}, \boldsymbol{y}_{\leq T}) = \prod_t \left[ p_\theta(\boldsymbol{x}_t|\boldsymbol{x}_{<t}, \boldsymbol{z}_{\leq t}, \boldsymbol{y}_{\leq t}) p_\theta(\boldsymbol{z}_t|\boldsymbol{x}_{<t}, \boldsymbol{z}_{<t}, \boldsymbol{y}_{<t}) p_\theta(\boldsymbol{y}_t|\boldsymbol{x}_{<t}, \boldsymbol{z}_{<t}, \boldsymbol{y}_{<t}) \right],$$

with generating distribution $p_\theta(\boldsymbol{x}_t|\boldsymbol{x}_{<t}, \boldsymbol{z}_{\leq t}, \boldsymbol{y}_{\leq t})$ and prior distributions $p_\theta(\boldsymbol{z}_t|\boldsymbol{x}_{<t}, \boldsymbol{z}_{<t}, \boldsymbol{y}_{<t})$ and $p_\theta(\boldsymbol{y}_t|\boldsymbol{x}_{<t}, \boldsymbol{z}_{<t}, \boldsymbol{y}_{<t})$ for latents $\boldsymbol{z}_t$ and $\boldsymbol{y}_t$. Note that in our sequential setting, the priors are included in the optimization rather than being represented by fixed distributions to account for latent dynamics and effective sampling at inference time.

To derive the variational distributions, we differentiate between observed labels $\boldsymbol{y}$ and latent variables $\tilde{\boldsymbol{y}}$ in the remainder. Using the factorization introduced in Section 3.1 yields

$$q_\phi(\boldsymbol{z}_{\leq T}|\boldsymbol{x}_{\leq T}, \boldsymbol{y}_{\leq T}) = \prod_t q_\phi(\boldsymbol{z}_t|\boldsymbol{x}_{\leq t}, \boldsymbol{z}_{<t}, \boldsymbol{y}_{\leq t}) \text{ and } q_\phi(\tilde{\boldsymbol{y}}_{\leq T}|\boldsymbol{x}_{\leq T}) = \prod_t q_\phi(\tilde{y}_t|\boldsymbol{x}_{\leq t}, \boldsymbol{z}_{<t}, \boldsymbol{y}_{<t}).$$
$$(2)$$

Labeled instances render reasoning over latents $\tilde{\boldsymbol{y}}_{\leq T}$ unnecessary since the true $\boldsymbol{y}_{\leq T}$ is already known. Hence, the right-hand side in Eqn (2) can be ignored when sampling from $\mathcal{D}_L$. We arrive at the following results and refer to Appendix **??** for the proofs.

**Theorem 1.** *A lower bound on* $\log p_\theta(\boldsymbol{x}_{\leq T}, \boldsymbol{y}_{\leq T})$ *in Eqn (1) is given by*

$$\log p_\theta(\boldsymbol{x}_{\leq T}, \boldsymbol{y}_{\leq T}) \geq \sum_t \log p_\theta(y_t|\boldsymbol{x}_{<t}, \boldsymbol{z}_{<t}, \boldsymbol{y}_{<t}) + \mathbb{E}_{q_\phi(\boldsymbol{z}_t|\boldsymbol{x}_{\leq t}, \boldsymbol{z}_{<t}, \boldsymbol{y}_{\leq t})} \left[ \log p_\theta(\boldsymbol{x}_t|\boldsymbol{x}_{<t}, , \boldsymbol{z}_{\leq t}, \boldsymbol{y}_{\leq t}) \right]$$

$$- \mathcal{KL}[q_\phi(\boldsymbol{z}_t|\boldsymbol{x}_{\leq t}, \boldsymbol{z}_{<t}, \boldsymbol{y}_{\leq t}) \parallel p_\theta(\boldsymbol{z}_t|\boldsymbol{x}_{<t}, \boldsymbol{z}_{<t}, \boldsymbol{y}_{<t})] \equiv \sum_{t=1}^T -\mathcal{J}_{S-MAT}(\boldsymbol{x}_t, \boldsymbol{y}_t). \quad (3)$$

Correspondingly, for unlabeled instances drawn from $\mathcal{D}_U$, sequential label information need to be estimated as shown in the following theorem.

**Theorem 2.** *Let* $\mathcal{H}(\beta)$ *be the entropy of quantity* $\beta$. *A lower bound on* $\log p_\theta(\boldsymbol{x}_{\leq T})$ *in Eqn (1) is given by*

$$\log p_\theta(\boldsymbol{x}_{\leq T}) \geq \sum_t \left( \mathcal{H}\big(q_\phi(\tilde{\boldsymbol{y}}_t|\boldsymbol{x}_{\leq t}, \boldsymbol{z}_{<t}, \boldsymbol{y}_{<t})\big) - \mathbb{E}_{q_\phi(\tilde{\boldsymbol{y}}_t|\boldsymbol{x}_{\leq t}, \boldsymbol{z}_{<t}, \tilde{\boldsymbol{y}}_{<t})} \big[ \mathcal{J}_{S-MAT}(\boldsymbol{x}_t, \tilde{\boldsymbol{y}}_t) \big] \right)$$

$$\equiv \sum_t -\mathcal{J}_{U-MAT}(\boldsymbol{x}_t). \quad (4)$$

Eqn (4) encodes the behavior of object $q_\phi(\boldsymbol{y}_{\leq T}|\boldsymbol{x}_{\leq T})$ when dealing with unsupervised data. Since data likelihood $\log p_\theta(\boldsymbol{x}_t|\boldsymbol{x}_{<t}, , \boldsymbol{z}_{\leq t}, y_{\leq t})$ tends to comprise the largest factor in $\mathcal{L}(\boldsymbol{x}_t, y_t)$, taking the expectation wrt. $q_\phi(\tilde{\boldsymbol{y}}_t|\boldsymbol{x}_{\leq t}, \boldsymbol{z}_{<t}, \tilde{\boldsymbol{y}}_{<t})$ encourages the model to realize the highest probability mass for the latent label valuel $y \in \mathcal{Y}$ that incurs the smallest reconstruction loss compared to other candidates in label space. This is a desirable property that reflects our assumption of data generation.

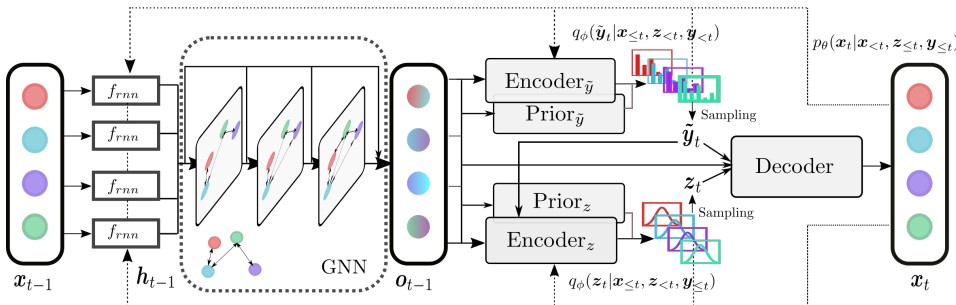

Figure 2: The full computational logic when processing unobserved multi-agent segments. Given temporal context encoded using an RNN module, the model leverages a customized GNN architecture based on [52, 53] (Section 3.5) to infer posteriors (and priors) over the discrete ($\boldsymbol{y}_t$) and continuous ($\boldsymbol{z}_t$) latent factors. The realized latent values, in conjunction with the GNN output, serve as input for the decoding module, which generates distribution over agent movements.

The full evidence lower bound (ELBO) is obtained by combining the lower bounds for all data $\mathcal{D}$,

$$p(\mathcal{D}) \geq \sum_{\mathcal{D}_L} \sum_t -\mathcal{J}_{S-MAT}(\boldsymbol{x}_t, \boldsymbol{y}_t) + \sum_{\mathcal{D}_U} \sum_{t=1}^T -\mathcal{J}_{U-MAT}(\boldsymbol{x}_t). \tag{5}$$

This formalization focuses on either fully labeled or unlabeled observations. However, a more general formulation can be obtained to also allow for partially labeled data points.

### 3.4 Extension to Classification Settings

Intrinsic to our contribution is an encoding module that argues over the label space, $q_\phi(\boldsymbol{y}_{\leq T}|\boldsymbol{x}_{\leq T})$, and hence can be used as a classification network annotating new observations. However, the ELBO in Eqn (5) is oblivious to classification and simply ignores relevant gradient updates for $\mathcal{J}_{S-MAT}(\boldsymbol{x}_t, \boldsymbol{y}_t)$ by discarding Eqn (2). This is clearly inappropriate for semi-supervised learning where the overall goal is to learn an unknown mapping $\boldsymbol{x} \mapsto \boldsymbol{y}$. Following [26], we circumvent this by heuristically incorporating Eqn (2) into the label-dependent objective. Hence, the full training criterion to learn $\{\theta, \phi\}$ is given by

$$\mathcal{J}_{MAT}(\theta, \phi; \mathcal{D}) = \lambda_0 \sum_{\mathcal{D}_U} \sum_t \mathcal{J}_{U-MAT}(\boldsymbol{x}_t)$$
$$+ \sum_{\mathcal{D}_L} \sum_t \left( \mathcal{J}_{S-MAT}(\boldsymbol{x}_t, \boldsymbol{y}_t) - \lambda_1 \log q_\phi(y_t|\boldsymbol{x}_{\leq t}, \boldsymbol{z}_{<t}, \boldsymbol{y}_{<t}) \right), \tag{6}$$

where $\lambda_1$ balances generative and discriminative learning and $\lambda_0$ the contribution of labeled and unlabeled data, respectively. Thus, $\mathcal{J}_{MAT}$ contains fully (un)supervised modeling tasks as special cases via adjusting $\lambda_1, \lambda_2$ accordingly. Note that the resulting supervised part in Eqn 6 (right part) differs from its unsupervised counterpart only in observing $\boldsymbol{y}_t$ instead of factorizing over $\mathcal{Y}$, which results in computing the log-likelihood (auxiliary loss) $\log q_\phi(y_t|\boldsymbol{x}_{\leq t}, \boldsymbol{z}_{<t}, \boldsymbol{y}_{<t})$ instead of the entropy $\mathcal{H}(q_\phi(\boldsymbol{y}_t|\boldsymbol{x}_{\leq t}, \boldsymbol{z}_{<t}, \boldsymbol{y}_{<t}))$.

### 3.5 Design Choices

As described in Section 3.2, we aim to learn a distribution over possible sequences of agent sets that may differ in share of discrete annotations (and cardinality). However, the previously derived objective functions only account for temporal dependencies, so that independence of the agent dimension is required to realize permutation-invariant models. This assumption is trivially inappropriate for interactive systems where future agent movements need to be coordinated with other agents. Thus, it is common to propose some form of graph encoding strategy [46] to account for both, equivariance and agent interactions, respectively.

Intuitively, optimal interaction modeling allows to adaptively accommodate structural information from different levels of granularity for each agent. In team sports, for example, agent nodes should

be able to capture immediate player influences as well as holistic team strategies dependent on the interaction structure best suited for the task at-hand. Motivated by the previous considerations, we customize an GNN architecture that aims to mimic the desired behavior. Specifically, we stack attention-based GNN layers [52] with particularly designed skip-connections [53] and construct the assumed graph structure using the $k$ (spatially) closest agents.

There are various possibilities to incorporate graph-based approaches into the overall scheme[1]. An efficient way to model the distributions defining $\mathcal{J}_{MAT}$ is via hidden agent states $\boldsymbol{h}_t^{(a)}$ conditioned on representations $\boldsymbol{o}_{t-1}^{(a)}$,

$$\boldsymbol{h}_t^{(a)} = f_{rnn}(\boldsymbol{x}_t^{(a)}, \boldsymbol{z}_t^{(a)}, \boldsymbol{y}_t^{(a)}, \boldsymbol{o}_{t-1}^{(a)}) \quad \text{with} \quad \boldsymbol{o}_t = \text{GNN}(\boldsymbol{h}_t),$$

where $f_{rnn}$ denotes an RNN transition function, GNN is the described GAT-based GNN and $\boldsymbol{o}_t = \{\boldsymbol{o}_t^{(a)} \mid a \in \mathcal{A}\}$ refers to the set of updated agent representations[2]. Since the elements in set $\boldsymbol{o}_t$ aggregate neighboring information of the RNN outputs $\boldsymbol{h}_t$ via graph networks, the inferred feature vectors describe the entire interactive past of individual agents. While it is reasonable to additionally enforce intra-timestep dependencies on the latent variables using encoding and decoding GNN modules, we argue that capturing past spatiotemporal patterns is sufficiently informative for high-frequency data [43]. Hence, assuming conditional independence within $t$, the joint movement distribution factorizes across the agent dimension and is given by

$$p_\theta(\boldsymbol{x}_t|\boldsymbol{x}_{<t}, \boldsymbol{z}_{\leq T}, \boldsymbol{y}_{\leq T}) = p_\theta(\boldsymbol{x}_t|\psi_t) = \prod_{a \in \mathcal{A}} p_\theta(\boldsymbol{x}_t^{(a)}|\psi_t^{(a)})$$

where $\psi_t = \{\psi_t^{(a)} \mid a \in \mathcal{A}\}$ are the distribution parameters and $\psi_t^{(a)} = \varphi(\boldsymbol{z}_t^{(a)}, \boldsymbol{y}_t^{(a)}, \boldsymbol{o}_{t-1}^{(a)})$. The computational logic of the full architecture is schematically depicted in Fig. 1 and 2.

## 4 Empirical Evaluation

For evaluation, we focus on team sports as the coordination of players renders these tasks more difficult than other domains [34].[3] Hence, we experiment on STATS SportVU NBA[4] for comparison, and tracking data from soccer games of the german national team. As detailed in Section 3.2, we use agent velocities as input to all models and assume linear motion between consecutive observations. For NBA, we adopt the experimental setup and processing strategy from [38][5].

The STATS SportVU NBA data comprises tracking positions of offensive plays from the 2016 NBA regular season covering more than 1200 different games where game segment are given by sequences of length 50 and contain two-dimensional positions of all agents (10 players and ball) sampled at 5 frames per second. The data is split into 60% training, 20% validation, and 20% test sets. All data is translated so that the origin of the underlying coordinate system is mapped onto the top-left corner.

The soccer data consists of 12 matches where positions of players and ball are sampled at 25 frames per second. The tracking data is accompanied by manual event annotations that we will also make use of in the remainder. Models are trained on eight matches, the remaining four are distributed evenly into validation and test data (two matches each).

### 4.1 Baselines

We evaluate versus several baselines on two distinct tasks: *trajectory forecasting* (Section 4.2 and 4.4) and *spatiotemporal classification* (Section 4.3 and 4.4). Since our approach constitutes the first semi-supervised model for label predictions in multi-agent scenarios, we need to rely on self-constructed supervised baselines for scenarios where only a few labels are available, which we discuss in more detail in the relevant sections.

---

[1] We empirically evaluate different configurations in Appendix **??**.

[2] Empirically, we observe performance gains when exclusively processing discriminative information $\boldsymbol{y}_{\leq T}$ using separate recurrent and graph networks. Appendix **??** outlines the implementation details.

[3] We report on results on Stanford Drone Data (SDD) [41] in the Appendix.

[4] https://github.com/linouk23/NBA-Player-Movements

[5] The source code is available at https://github.com/fassmeyer/MAT_NeurIPS22.

Table 1: Error for NBA in meters for a prediction interval of 10 timesteps with an observation period of 40 timesteps. Bold is the lowest avg $L_2$ & final $L_2$ for the supervised and unsupervised generative models, respectively.

|  | GVRNN | dNRI | GRIN | DAG-Net. | U-MAT | S-MAT |
|---|---|---|---|---|---|---|
| avg $L_2$ | 2.60 | 2.77 | 3.00 | 2.10 | *2.31* | **1.94** |
| final $L_2$ | 5.66 | 5.52 | 6.12 | 4.28 | *4.80* | **3.99** |

By contrast, trajectory forecasting is an active topic in different fields (see, e.g., [42] for an overview), which allows comparisons with a wide range of baseline models. It is difficult to determine the current state-of-the-art in generative methods for sports tracking data because there is no consistent setting in terms of forecasting horizon or data (see e.g., [32, 38]). While some approaches are restricted to making predictions for a predetermined horizon, our method learns spatiotemporal representations autoregressively and thus can be used for the full range of prediction scenarios. To include a variety of methods, we conduct two sets of experiments over short (10 timesteps) and long (40 timesteps) prediction horizons. We benchmark against the most recent and methodologically important work described below.

Fully supervised generative models such as *DAG-Net* [38] and *Weak-Sup* [60] use weak labels in the form of agent objectives (heuristically inferred prior to model training) to improve trajectory prediction. In these settings, an agent's objective at each timestep $t$ is to reach the next area where the observed movement speed falls below a predefined threshold. Adopting the experimental framework of [38] allows us to make further comparisons with *Social-Ways* [2] and *STGAT* [22] without additional experiments. STGAT [22] augments a standard GAT with an LSTM [20] to capture both spatial and temporal dependencies of social interactions while Social-Ways uses Info-GAN [6] to learn multimodal predictive distributions.

The majority of generative models for sequential multiagent systems operate in a fully-unsupervised fashion. *GVRNN* [55] is a graph extension of the VRNN for explicitly modeling joint agent movements for basketball and soccer data. *dNRI* [17] dynamically extracts interaction types that determine the parametrization of the decoder. Finally, *GRIN* [32] recovers (static) interactions via attention using a disentangled latent space.

## 4.2 Exploring Boundary Cases for Trajectory Prediction

**Methods** We study the purely (un)supervised generation of trajectories by maximizing the respective lower bounds on NBA. For the fully-supervised variant, the weak labels described in Section 4.1 can be naturally integrated into the overall scheme via treating them as semantic concepts $y_t^{(a)}$. These target areas naturally vary over time so that the sequential label $\boldsymbol{y}_{\leq T}$ always denotes the next desired position for every involved agent at time $t$. Following related work [38, 60], we obtain 90 different areas/class labels. The underlying data distribution is estimated by maximizing Eqn (3). The supervised variant of our contribution is denoted by *S-MAT*.

Although done in [38, 60], direct comparison of unsupervised and supervised generative models is inappropriate inasmuch as the latter use ground-truth labels over the entire observation period (which include future information) at prediction time. Thus, we additionally benchmark against a fully unsupervised instantiation of our proposed framework lacking ground-truth information for structuring the discrete latent subspace. To encourage the model to learn semantically meaningful concepts describing distinct movement patterns, we parametrize the decoder dependent on the predicted agent class $\tilde{y}_t^{(a)}$. In this context, $\tilde{y}_t^{(a)}$ may be best thought of dynamically assigned agent roles that accurately explain the observed trajectories. We maximize Eqn (4) using 3 agent types and refer to the unsupervised variant as *U-MAT*. Further details for S-MAT and U-MAT are given in Appendix **??**.

**Metrics** We measure the quality of the learned multimodal distribution using common standard metrics: the minimum over 20 generated samples of the *average* and *final* $L_2$ distance between predicted $\hat{\boldsymbol{x}}_{\leq T}^{(a)}$ and observed positions $\boldsymbol{x}_{\leq T}^{(a)}$.

Table 2: Error for NBA in meters for a prediction interval of 40 timesteps with an observation period of 10 timesteps. Top rows show results for offense, bottom rows the defense. Bold is the lowest avg $L_2$ & final $L_2$ for the supervised and unsupervised generative models, respectively.

|  | STGAT | Social-Ways | GVRNN | Weak-Sup. | DAG-Net | U-MAT | S-MAT |
|---|---|---|---|---|---|---|---|
| avg $L_2$ | 9.94 | 9.91 | 9.73 | 9.47 | 8.98 | *9.01* | **8.11** |
| final $L_2$ | 15.80 | 15.19 | 15.89 | 16.98 | 14.08 | *13.28* | **12.52** |
| avg $L_2$ | 7.26 | 7.31 | 7.29 | 7.05 | 6.87 | *6.88* | **6.21** |
| final $L_2$ | 11.28 | 10.21 | 10.62 | 10.56 | 9.76 | *9.04* | **8.45** |

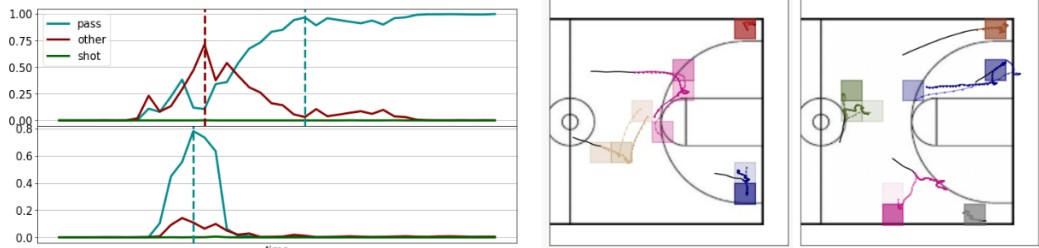

Figure 3: *Left:* Prediction probabilities for different events on soccer data. The dashed lines correspond to the human annotations. *Right*: Exemplary rollouts for offensive players on basketball data. The figure shows observed (light color) and generated player trajectories (intense color) as well as label information indicated by colored boxes, where the color intensity corresponds to the ground-truth frequency of the (weakly obtained) location-based labels.

**Results**   Tables 1 and 2 show the results. The proposed approaches clearly outperform their peers and realize the lowest average $L_2$ and final $L_2$ distances. The *MAT* variants effectively accommodate not only mutual influences across agents, but also discrete generative factors, leading to better approximations of the underlying multi-modal distributions. Unsurprisingly, Table 1 also shows that defending players exhibit more structure and are easier to predict than offensive players. The results of an ablation study are contained in Appendix **??**. The right part of Fig. 3 displays exemplary rollouts that underline models' ability to capture highly complex changes in movement directions. The results indicate that the proposed framework denotes a valuable contribution to the large body of research that explicitly addresses modeling multi-agent data accurately.

### 4.3   Semi-Supervised Classification

We now extend the experimental protocol by targeting accurate label discovery in semi-supervised scenarios. We refer to the full model simply as *MAT* where $\lambda_0, \lambda_1 > 0$ and incorporate the fully supervised *S-MAT* ($\lambda_0 = 0$) as an additional baseline. We also compare to an RNN-based classification network that addresses inter-agent dependencies by inferring hidden states of agents by the GNN architecture described in Section 3.5. A final softmax layer is used for classification, and the model is optimized by minimizing the negative log-likelihood. We refer to this baseline as *MA-RNN*.

Figure 4 shows the performance of the models on NBA when using $50\%$ labeled and $50\%$ unlabeled examples (left side) and when using varying portions of labeled data (right side), while the supervised baselines have access only to the labeled part. The semi-supervised *MAT* clearly emerges as the strongest classifier as it converges to the lowest generalization error during a training run and the performance benefit increases via reduction of labeled data. Comparing the numbers to its supervised peer (*S-MAT*) highlights the value of incorporating unlabeled data that is effectively utilized by *MAT* to inform classification decisions. Comparing the fully supervised approaches *S-MAT* and *MA-RNN*, the importance of an effective regularization mechanism becomes obvious; *MA-RNN* quickly overfits while the lower-bound in *S-MAT* acts as a regularizer and guarantees gradually increasing predictive accuracies on validation sets.

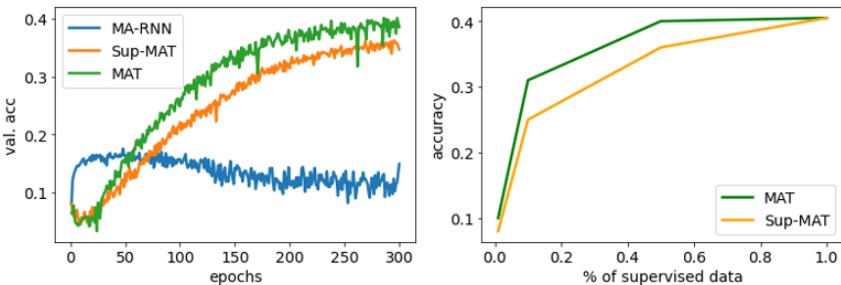

Figure 4: Semi-supervised results on NBA. *Right*: Progression of accuracy during a training run. *Left*: Test accuracies for varying amount of labeled data.

Table 3: Results on soccer data. *Left*: Semi-supervised data generation for a prediction interval of 10 timesteps with an observation period of 40 timesteps (errors in meters). *Right*: Semi-supervised event detection.

| NAME | AVG $L_2$ | FINAL $L_2$ |
| --- | --- | --- |
| VRNN | 2.91 | 7.10 |
| VRNN + GNN | 2.87 | 6.98 |
| GVRNN | 2.89 | 6.97 |
| S-MAT | 2.87 | 6.91 |
| MAT-DIAG | 2.84 | 6.90 |
| MAT | **2.81** | **6.88** |

| NAME | ACCURACY | F1 |
| --- | --- | --- |
| MA-RNN-DIAG | 0.74 | 0.80 |
| MA-RNN | 0.85 | 0.88 |
| S-MAT | 0.82 | 0.85 |
| MAT | **0.88** | **0.92** |

## 4.4 Combining Generation and Classification in Semi-Supervised Settings

In our last set of experiments, we study the combination of trajectory generation and classification and resort to tracking data from elite soccer. The reason for favoring this data over NBA is that the soccer tracking data is manually annotated by experts from the data provider. We focus on labels $\mathcal{Y} = \{pass, other\ ball\ action, shot, none\}$, where class *none* denotes the absence of all other labels in a frame and use 20% annotated data. Every label is propagated to the previous five and the subsequent 30 frames. We generate a balanced training set where half of the segments carry label *none*.

The main purpose of this set of experiments is to quantitatively validate the benefits of including expensive human annotations in the proposed modeling framework for structuring the discrete latent subspace. This approach has low additional costs because the method operates semi-supervised, so only a small subset of multi-agent segments needs to be annotated. Given the higher sampling rate and less interactive nature of soccer dynamics, we also aim to evaluate the extent to which social dependency modeling modules achieve empirical benefits. We design our baselines accordingly.

**Generation** For trajectory prediction, we compare *MAT* to *VRNN* [8], an interactive version of thereof with GNNs on the hidden states (*VRNN+GNN*), *GVRNN* [55], *S-MAT* with long-term goals (introduced in Section 4.2), and a diagonal MAT version, *MAT-Diag*. Note that the *VRNN* constitutes a strong competitor for the task-at-hand, as it effectively corresponds to a diagonal GVRNN, which has been shown to produce competitive results on soccer data [55].

The left part of Table 3 shows the results. Although the results are generally more similar compared to the basketball experiments, the semi-supervised *MAT* generates trajectories that are closest in both metrics to the observed ones. We hypothesize that decreasing the frequency of the data and extracting multi-agent segments that merely consist of highly interactive plays yields an significant increase in performance gap.

**Classification** *MAT* learns a model over the label space $\mathcal{Y}$ simultaneously to the generative model. To evaluate classification performance, we turn the output of the corresponding softmax for a given frame into a class label whenever it exceeded a predefined threshold. We compare the prediction performance to the classification baselines introduced in Section 4.4.

The resulting accuracies and F1 scores are summarized in Table 3. Once more, *MAT* clearly beats the baselines and stands out by the highest accuracy and F1 scores. The result impressively demonstrates the benefit of including unlabeled data for the task. Surprisingly, *MA-RNN* performs better than *S-MAT*. This finding is in contrast to the results on NBA shown in Figure 4. However, soccer is played on a much larger pitch and movements are likely more linear compared to basketball, which leads to less multimodality of the distributions involved and a consequently decreased benefit of variational methods. Together with a significantly reduced label space and sufficient amounts of data this may lead to a simpler learning task where *MA-RNN* is less prone to overfitting.

The left part of Fig. 3 shows exemplary prediction probabilities for two segments and three possible events. In both segments, the algorithm is highly confident no shot-action will occur in the near future. The segment on top contains an event of the class *other ball action* followed by an event of class *pass*. Both are correctly identified by *MAT*. The former is clearly indicated by a peak in the corresponding probability chart, which then decreases to give rise to the following *pass* action. The segment on the bottom shows solely a single *pass* event that is also clearly identifiable by a peak at the correct frame.

## 5   Conclusion

We presented semi-supervised variational autoencoders for spatiotemporal multi-agent scenarios. The proposed approach effectively combined ideas from semi-supervised variational autoencoders, variational recurrent neural networks, and graph neural networks. Empirically, our approach clearly outperformed previous state-of-the-art models in interactive sequential generation and (semi-supervised) classification tasks in all experiments. The performance underlines the benefit of including unlabeled data in spatiotemporal problems where labeled sequences are either scarce or assembled from weak makeshift signals.

**Potential societal impact**   We have a clear focus on team sports. Though dual use of the proposed technique is certainly possible (e.g., military domains), it may be a bit far fetched as we have no expertise in these domains and would only be able to deliver uneducated guesses about whether this is realistic or not. Generally, the data and tasks at-hand are privacy-sensitive as the targets are human individuals. While the soccer and basketball players are probably identifiable in the data, they/their contract/their clubs agreed in recording the data.

## 6   Acknowledgments

We would like to thank the German Football Association (DFB) for providing data for this study.

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
