# OpenReview forum: "Semi-Supervised Generative Models for Multiagent Trajectories"
_NeurIPS.cc/2022/Conference — NeurIPS 2022 Accept_

### Official Review · Reviewer_CLhb · 2022-07-10

**Rating:** 5
**Confidence:** 3
**Soundness:** 3 good
**Presentation:** 2 fair
**Contribution:** 2 fair

**Summary:**

This work presents a new semi-supervised approach for multi-agent trajectory forecasting and classification. In particular, a new factorization and loss formulation is presented that handles both unsupervised and supervised data simultaneously in the same framework. The resulting model was evaluated on two sports trajectory datasets (NBA and soccer), demonstrating improvements over a variety of baselines.

**Questions:**

- What are the core contributions of this work? Is it Equation (1)? Is it an architectural improvement upon an existing neural network architecture? How do the contributions build upon existing works?

- How does the method's performance compare to others on physically-grounded units such as meters?

- Do any of the other baseline methods provide probabilistic outputs? If so, it would be best to compare to them on probabilistic metrics such as negative log-likelihood, since minADE/minFDE are only approximations of probabilistic metrics. If it is not possible to directly compute likelihoods, what about comparing to other methods using a metric like the KDE-based NLL from https://arxiv.org/abs/1810.05993 which more closely approximates a probabilistic metric?


**Limitations:**

This work does not state any limitations or potential negative societal impacts of the method.

**Strengths And Weaknesses:**

### Strengths:

The method tackles an important problem and presents a sensible solution combining weakly-labelled data with unsupervised data.

It is difficult to judge if the method's formulation is novel, but if it is, then this method presents a novel formulation which combines two otherwise-disparate approach types.

The method is evaluated on real-world data comprised of interesting multi-human interaction scenarios.

### Weaknesses:

The abstract is quite light on grounded examples whose inclusion may better allow readers to appreciate this work and its applicability. For instance, what are some spatiotemporal patterns that users wish to identify? What kinds of annotations are required for situations? Are they the positions of agents or text descriptions of events occurring in a scene? These are stated later in the paper, but it would be helpful to state the examples upfront in the abstract as well.

It is quite difficult to identify concretely the core contributions of this work. Section 3 (Main Contribution) spans 3+ pages and interweaves background material (e.g., Section 3.1) with the description of the method at hand. Is the core contribution Equation (1) and its subsequent term derivations?

In Section 3.3, the authors frequently refer to the dependency structures/factorization introduced in Section 3.1, however it is difficult to quickly parse exactly which dependency structures were introduced from text (presumably this is referring to the factorization of p(x, z, y) and q(y, z | x)). Perhaps a figure in Section 3.1 illustrating the probabilistic graphical model would help readers more?

Assuming that the number of agents A remains constant across time (Line 174-175) limits the applicability of this work to many real-world domains such as autonomous driving or social navigation, where the number of detected agents can change as they drift in and out of sensor range or visibility.

Lines 212-215, the soccer dataset split has an error somewhere: "The soccer data consists of 12 matches ... Models are trained on 8 matches, the remaining 6 are ...". Are there 12 total matches or 14?

Line 243: which related work was followed?

Table 1 and 2 seem strange to include together, one shows a prediction horizon of 10 timesteps vs the other's 40 timesteps, with no other changes apparent (other than the methods compared against). Accordingly, it does not appear that Table 1 provides much value compared to Table 2 which focuses on a longer time horizon as well as compares offensive vs defensive player prediction accuracy.

Line 260: Why are all metrics reported on un-normalized scales? Why not meters?

In Section 4.2, do any of the other methods provide probabilistic predictions? It would be best to compare to them on metrics such as negative log-likelihood if so, since minADE/minFDE are only approximations of probabilistic metrics. If not possible to directly compute likelihoods, why not compare using something like the KDE-based NLL from https://arxiv.org/abs/1810.05993?

The left part of Table 3 is hard to interpret, since all methods lie within ~0.10 units of performance of each other. Is this a lot or a little? Without physically-grounded units, it is difficult to tell.

### Minor Comments

The title does not match the OpenReview title.

Fig. 1 is referenced 2 pages away from where it is graphed, perhaps moving it closer to Line 200 would be better?

---

> ### Author Response · Authors · 2022-08-02
> **Response to Reviewer CLhb**
>
> We thank the reviewer for the detailed review as well as the suggestions for improvement. We hope the following response addresses your concerns.
>
> *The abstract is quite light on grounded examples whose inclusion may better allow readers to appreciate this work and its applicability. For instance, what are some spatiotemporal patterns that users wish to identify? What kinds of annotations are required for situations? Are they the positions of agents or text descriptions of events occurring in a scene? These are stated later in the paper, but it would be helpful to state the examples upfront in the abstract as well.*
>
> We thank you for this suggestion. We also believe that a more detailed description of the concrete applications and contributions already in the abstract would increase readership interest in this paper. Our new abstract is as follows (cf. also revised manuscript):
>
> Analyzing the spatiotemporal behavior of multiple agents is of great interest to many communities. Existing probabilistic models in this regime employ either unsupervised generative settings, in which the latent space is described fully by discrete or continuous representations, or, are alternatively formalized in a (fully) supervised framework where weakly preserved labels add explicit information to a continuous latent representation learned from the data. To overcome the resulting limitations, we propose a novel objective function for processing multi-agent trajectories based on semi-supervised variational autoencoders, where equivariance and interaction of agents are modeled via costumized graph networks. Our formulation disentangles discrete and continuous effects and allows discrete behavioral indicators in arbitrary quantity and annotation type to guide the generation process. This lifts applicability to relevant prediction problems beyond the generation of collective movements and provides an effective solution to incoporate expensive domain knowledge into interactive multi-agent systems. Empirically, our model outperforms various state-of-the-art baselines in generating future agent movements on interactive real-world datasets. We also show that our approach effectively learns to leverage unsupervised multi-agent sequences to improve classification of long-term locations as well as manually annotated situations on sports tracking data.
>
> *In Section 3.3, the authors frequently refer to the dependency structures/factorization introduced in Section 3.1, however it is difficult to quickly parse exactly which dependency structures were introduced from text (presumably this is referring to the factorization of p(x, z, y) and q(y, z | x)). Perhaps a figure in Section 3.1 illustrating the probabilistic graphical model would help readers more?*
>
> Yes, we refer here to the dependency structures of the generation and inference parts of semi-supervised VAEs introduced in Section 3.1. To demonstrate the essential procedures of the prediction framework, we added a probabilistic graphical model shown in Figure 1 (right part) (see also link below). This illustration also contains the factorization of $p(x, z, y)$ and $q(y, z | x)$ and we added a pointer to the figure in the text of Section 3.1. We also added more details in the original architecture visualization (Figure 1, left) to improve clarity.
>
> **Note:** The updated Figure can be also accessed through the following anonymous link: [https://github.com/hghfghfhhdc/neurips_rebuttal/blob/main/updated_architecture.png](https://github.com/hghfghfhhdc/neurips_rebuttal/blob/main/updated_architecture.png)
>
>
> *Assuming that the number of agents A remains constant across time (Line 174-175) limits the applicability of this work to many real-world domains such as autonomous driving or social navigation, where the number of detected agents can change as they drift in and out of sensor range or visibility.*
>
> Our method has no limitations in terms of applicability to scenes with dynamic cardinality of agent sets (cf. drone experiments in Appendix C) compared to prior work in this field. This restriction occurs in approaches with separate models for each agent, such as Weak-Sup [63], that maintains a separate VRNN [6] per agent and thus cannot be included as a baseline in the drone experiments. We agree that lines 174-175 are misleading in this respect (which we initially included due to our focus on sports tracking data), so we have omitted the sentence in the revised version and made this more clear in the text of Section 3.5.. In fact, compared to existing work on social navigation or self-driving cars (e.g., [37, 38, 32, 48, 49, 1, 21]), our method is more general as it allows to incorporate expensive domain knowledge into interactive multi-agent systems, which could benefit the above application domains in many ways.

---

> > ### Author Response · Authors · 2022-08-02
> > **Response to Reviewer CLhb (cont.)**
> >
> > *It is quite difficult to identify concretely the core contributions of this work. Section 3 (Main Contribution) spans 3+ pages and interweaves background material (e.g., Section 3.1) with the description of the method at hand. Is the core contribution Equation (1) and its subsequent term derivations?*
> >
> > Yes, the new problem formulation and the derived theorems constitute the key contribution of this work. Consequently, the proposed architecture designed based on the inferred objective function denotes a novel algorithm in the spatiotemporal regime (though the methodological tools employed has been used before). To clarify the relationship to [27] and to highlight our novelities, we have added the right part to Figure 1 and updated our architecture visualization (left part, Figure 1), extended its caption, made general clarifying edits on the novelities/differences throughout the paper (Abstract; Sections 1, 2 and 3), and altered section titles (cf. Section 3, as you suggested in your comment).
> >
> > To summarize, our work is distinct from others in the following ways:
> >
> > - We formalize a probabilistic generative model in spatiotemporal regimes by a semi-supervised approach. This allows us to
> >     - Model simultaneously disentangled discrete (defined by the user of the system, e.g., expert knowledge) and continuous effects. Previous VAE-based spatiotemporal models employ either unsupervised generative settings, in which the latent space is described fully by discrete [13, 49, 31, 28, 16, 14] or continuous [28, 52, 59, 16, 14, 33] variables, or, are alternatively formalized in a supervised framework where weakly preserved ``macro-intents" act as explicitly defined discrete information while the continuous latent subspace is learned from data [64, 65, 40]. At a high level, our contribution subsumes these approaches into a unified framework via a disentangled latent space that allows (discrete) supervision signals (or more generally, behavioral indicators) in arbitrary quantity and annotation type to govern the generative process. Intuitively, this increases the "scope" of latent information captured by the method and improves upon existing SOTA methods in trajectory forecasting settings.
> >     - Lift applicability to highly relevant prediction problems beyond representational or generative modeling. For example, intrinsic to our approach is an encoding module that argues over the label space and can thus be used in real-world settings with expensive manual annotations. We validated the advantages empircally. Existing generative models for spatiotemporal pattern detection focus almost exlusively on mulit-agent trajectory prediction.
> > - Though the methodological tools (RNN+GNN) have been used before, the concrete architectures that emerge based on the derived theorems and practical considerations (cf. e.g., Section 3.5.) are novel. The architectual design choices are validated empirically via outperforming SOTA baselines (Section 4, Appendix D) and ablation studies (Appendix E).
> >
> > *Lines 212-215, the soccer dataset split has an error somewhere: "The soccer data consists of 12 matches ... Models are trained on 8 matches, the remaining 6 are ...". Are there 12 total matches or 14?*
> >
> > Thanks! In total, we had 12 matches with 2 games for validation and testing. We corrected this in the revised version.
> >
> > *Table 1 and 2 seem strange to include together, one shows a prediction horizon of 10 timesteps vs the other's 40 timesteps, with no other changes apparent (other than the methods compared against). Accordingly, it does not appear that Table 1 provides much value compared to Table 2 which focuses on a longer time horizon as well as compares offensive vs defensive player prediction accuracy.*
> >
> > As described in Section 4.1, we aim to compare our method with a variety of SOTA baselines for sports tracking data. However, some of these methods are limited in their flexibility in terms of observation and prediction periods, which led us to perform two sets of experiments (Table 1 and Table 2). For example, the source code of GRIN [33] is limited to setting T_obs=40 and T_pred=10 because the training method performs motion prediction by using a single data point x_t every num_timesteps/pred_steps (T/T_pred) and then predicting T_pred time steps (cf. their publically available source code). In addition, dNRI [16] only uses the observation period T_obs for model training, so a longer prediction horizon (and resulting shorter observation time) would yield inferior results. This is one possible explanation why dNRI [16] has so far only been evaluated on rather short prediction horizons. On the other hand, GVRNN [59], DAG-Net [40], and our framework can be applied to the full range of prediction scenarios since these methods learn spatiotemporal representations autoregressively (and thus numbers are reported in both tables). We added a clarification to the baseline section (cf. last paragraph, Section 4.1, revised version).

---

> > > ### Author Response · Authors · 2022-08-02
> > > **Response to Reviewer CLhb (cont.)**
> > >
> > > *Line 260: Why are all metrics reported on un-normalized scales? Why not meters?*
> > >
> > > As captioned in Table 1 and 2, we actually use meters as the evaluation criterion. Frequently, trajectory prediction methods center and normalize the trajectories to be in the range $\[-1,1\]$ in advance to model training. So with un-normalized scales, we refer to the original data format (meters in world coordinates). We agree that this expression can be confusing, so we clarified on the evaluation units in the revised version.
> > >
> > > I*n Section 4.2, do any of the other methods provide probabilistic predictions? It would be best to compare to them on metrics such as negative log-likelihood if so, since minADE/minFDE are only approximations of probabilistic metrics. If not possible to directly compute likelihoods, why not compare using something like the KDE-based NLL from [https://arxiv.org/abs/1810.05993](https://arxiv.org/abs/1810.05993)?*
> > >
> > > Since trajectory data is inherently non-determenistic and multimodal, all SOTA baselines (including the multi-agent literature in the related work section) constitute probabilistic approaches. However, in recent years, minADE/minFDE emerged as the metrics of choice to evaluate the learned distribution and to assess trajectory prediction capabilities. One potential reason for this development is that likelihoods do not necessary correspond to quality of generated samples (evicenced by e.g., [A] or [64]) . While minADE/minFDE are only approximations of probabilistic metrics, reporting them is often sufficient when quantitatively comparing generative models for sequential multi-agent data [62, 61, 32, 40, 37, 38] as NLL values usually do not provide additional insights.
> > >
> > > [A] Theis, Lucas, Aäron van den Oord, and Matthias Bethge. "A note on the evaluation of generative models." *arXiv preprint arXiv:1511.01844* (2015).
> > >
> > > *The left part of Table 3 is hard to interpret, since all methods lie within ~0.10 units of performance of each other. Is this a lot or a little? Without physically-grounded units, it is difficult to tell.*
> > >
> > > As stated earlier, we use meters for our metrics (indicated now in the caption of Table 3). Although the results are generally more similar compared to the basketball experiments, we would like to point out the following considerations:
> > >
> > > - For soccer dynamics, due to the bigger field and higher sampling rate (in our case, 25 Hz), players tend to run more linearly compared to basketball. Hence, we hypothesize that models with implicit independence assumptions across agents (and consequently no social interaction module), such as the VRNN baseline, tend to produce stronger results compared to basketball experiments since the past trajectory is frequently more informative for predicting its future than agent-agent dependencies (relatively speaking). This is also reflected in the comparatively small improvements when benchmarking interactive models against their (socially) independent peers (cf. VRNN vs. VRNN + GNN and MAT-Diag vs. MAT in Table 3).
> > > - As stated in line 295-297, the VRNN constitutes a strong competitor for the task at-hand since it effectively corresponds to a diagonal version of GVRNN [59] (referred to as GVRNN-Diag in [59]), which has been shown to produce competitive results on soccer data [59]. In fact, although the sampling rate is lower, the realized improvements of the best model in [59] are inferior to those in our work (3.11 vs. 3.07 for avg L_2, see Table 3 in [59]). We added GVRNN results to our manuscript in Table 3 (left side) which empircally validate this (VRNN: 2.91 vs. GVRNN: 2.89 vs. MAT (Ours): 2.81 for avg L_2).
> > > - As already indicated, considering the fairly high sampling rate and the resulting inherent lower prediction error, the observed improvements are actually reasonable. We hypothesize that decreasing the frequency of the data and extracting multi-agent segments that merely consist of highly interactive plays significantly yields an increase in performance gap.
> > >
> > > We added more details to the corresponding Section in the revised version.
> > >
> > > *The title does not match the OpenReview title.*
> > >
> > > We will ask the program chair to change the title of the paper to "Semi-Supervised Generative Models for Multi-Agent Trajectories" after acceptance (as suggested by R2).

---

> > > > ### Comment · Reviewer_CLhb · 2022-08-10
> > > > **Thank you for responding!**
> > > >
> > > > Thank you for responding to my comments, concerns, and suggestions! I do not have further questions at this time, and I appreciate your clarifications of my earlier misunderstandings about the work.
> > > >
> > > > After reviewing the rebuttal and other review comments/discussions, I will update my score positively. I also want to make this final comment in line with other reviewers: _Please_ make sure to proofread the work and correct typos in any original and modified text. For example, in the new abstract:
> > > >
> > > > - costumized -> customized
> > > >
> > > > - This lifts applicability -> This improves applicability (maybe another phrase altogether might be better?)
> > > >
> > > > - incoporate -> incorporate

---

### Official Review · Reviewer_iUsD · 2022-07-11

**Rating:** 6
**Confidence:** 3
**Soundness:** 3 good
**Presentation:** 3 good
**Contribution:** 3 good

**Summary:**

The work aims at learning temporal dynamics in a multi-agent setting from both labeled and unlabelled data. It combines the framework of VAE with semi-supervised learning and use a graph neural networks. Experiment results on two sports dataset shows superior performance of the proposed framework.

**Questions:**

1. Why is the U-MAT model in Table 1 called unsupervised? A predicted label is provided as input to the model. Calling it unsupervised may not be accurate.
2. Why different experiments settings/datasets are compared with different baseline models? For example in Table 1 and Table 2, different baseline models are compared against U-MAT and S-MAT but the experiment settings are very similar. In Table 3, VRNN does not appear in classification results. Even though VRNN is a sequential generative model, it’s hidden state can also be used for classification.


**Limitations:**

The work addresses the limitations of the work in terms of application in Section 4 but does not discusses potential negative social impact of the work. In the reviewer's opinion, discussing potential negative social impact may not be necessary for this particular paper as it is primarily a methodology work and the social impacts can depend on specific implication. However, it would definitely be welcome if the author can discuss the paper's limitations from broader social perspectives, in the rebuttal.

**Strengths And Weaknesses:**

Strength:
There are three major components in the proposed framework: VAE, semi-supervised learning and graph neural networks (GNNs). Each component is strongly motivated and they form a natural combination for the proposed problem setting of multi-agent trajectory modelling. I especially like the part of the application of semi-supervised learning to sequential data. It is of great practical potential. It can not only applied to sequential data without the need to labelling all of them which can be expensive but also provide the possibility of training on multiple sequential datasets including labeled ones and unlabeled ones. The paper is well structured with detailed introduction to the background knowledge and the work is easy to follow. The experiments are thorough. The work also compared against a rich set of baseline models on different tasks on NBA, soccer and Stanford drone dataset.

Weakness:
The novelty of the work is limited. The proposed approach is a combination of three existing approaches, sequential latent variable models, semi-supervised VAE and graph neural networks. Despite being a natural combination with strong motivation, the novelty of the work is  incremental. I would also encourage the author to include a separate paragraph that summarize the work’s contribution.

---

> ### Author Response · Authors · 2022-08-02
> **Response to Reviewer iUsD**
>
> We would like to thank you for your praise and constructive suggestions for improving the paper.
>
> *The novelty of the work is limited. The proposed approach is a combination of three existing approaches, sequential latent variable models, semi-supervised VAE and graph neural networks. Despite being a natural combination with strong motivation, the novelty of the work is incremental. I would also encourage the author to include a separate paragraph that summarize the work’s contribution.*
>
> Our contribution could be seen as a spatiotemporal generalization of [27]. However, this generalization is not straight forward but instead requires modeling two additional dimensions (time and space; see first paragraph of Section 2). To clarify the relationship to [27] and to highlight our novelities, we have added the right part to Figure 1 and updated our architecture visualization (left part, Figure 1), extended its caption, made general clarifying edits on the novelities/differences throughout the paper (Abstract; Sections 1, 2 and 3), and altered section titles (cf. Section 3).
>
> To summarize, our work is distinct from others in the following ways:
>
> - We formalize a probabilistic generative model in spatiotemporal regimes by a semi-supervised approach. This allows us to
>     - Model simultaneously disentangled discrete (defined by the user of the system, e.g., expert knowledge) and continuous effects. Previous VAE-based spatiotemporal models employ either unsupervised generative settings, in which the latent space is described fully by discrete [13, 49, 31, 28, 16, 14] or continuous [28, 52, 59, 16, 14, 33] variables, or, are alternatively formalized in a supervised framework where weakly preserved ``macro-intents" act as explicitly defined discrete information while the continuous latent subspace is learned from data [64, 65, 40]. At a high level, our contribution subsumes these approaches into a unified framework via a disentangled latent space that allows (discrete) supervision signals (or more generally, behavioral indicators) in arbitrary quantity and annotation type to govern the generative process. Intuitively, this increases the "scope" of latent information captured by the method and improves upon existing SOTA methods in trajectory forecasting settings.
>     - Lift applicability to highly relevant prediction problems beyond representational or generative modeling. For example, intrinsic to our approach is an encoding module that argues over the label space and can thus be used in real-world settings with expensive manual annotations. We validated the advantages empircally. Existing generative models for spatiotemporal pattern detection focus almost exlusively on mulit-agent trajectory prediction.
> - Though the methodological tools (RNN+GNN) have been used before, the concrete architectures that emerge based on the derived theorems and practical considerations (cf. e.g., Section 3.5.) are novel. The architectual design choices are validated empirically via outperforming SOTA baselines (Section 4, Appendix D) and ablation studies (Appendix E).
>
> **Note:** The updated Figure can be also accessed through the following anonymous link: [https://github.com/hghfghfhhdc/neurips_rebuttal/blob/main/updated_architecture.png](https://github.com/hghfghfhhdc/neurips_rebuttal/blob/main/updated_architecture.png)
>
> *Why is the U-MAT model in Table 1 called unsupervised? A predicted label is provided as input to the model. Calling it unsupervised may not be accurate.*
>
> In the fully unsupervised case (U-MAT), the model learns to structure the discrete (latent) subspace by itself without any annotation. From the predicted (latent) categorical distribution $q_\phi(\tilde{y}|.)$, we sample a label value and exploit separate motion predictors (different decoder parameterizations) $p_\theta(x_t|.)$ dependent on the realized y-value. Intuitively, this encourages the model to learn categories that describe fundamentally different movement patterns, which can be interpreted as dynamic “agent roles”. Thus, the concept of agent roles here is merely an intuitive explanation and realized via an inductive bias that increases the “scope” of latent information encoded without utilizing any supervision. To the best of our knowledge, parameterizing the generation module based on inferred agent categories is novel and could provide valuable insights for practitioners. We added more detailed model descriptions via Appendix C (cf. revised manuscript).

---

> > ### Author Response · Authors · 2022-08-02
> > **Response to Reviewer iUsD (cont.)**
> >
> > *Why different experiments settings/datasets are compared with different baseline models? For example in Table 1 and Table 2, different baseline models are compared against U-MAT and S-MAT but the experiment settings are very similar.*
> >
> > As described in Section 4.1, we aim to compare our method with a variety of SOTA baselines for sports tracking data. However, some of these methods are limited in their flexibility in terms of observation and prediction periods, which led us to perform two sets of experiments (Table 1 and Table 2). For example, the source code of GRIN [33] is limited to setting T_obs=40 and T_pred=10 because the training method performs motion prediction by using a single data point x_t every num_timesteps/pred_steps (T/T_pred) and then predicting T_pred time steps (cf. their publically available source code). In addition, dNRI [16] only uses the observation period T_obs for model training, so a longer prediction horizon (and resulting shorter observation time) would yield inferior results. This is one possible explanation why dNRI [16] has so far only been evaluated on rather short prediction horizons. On the other hand, GVRNN [59], DAG-Net [40], and our framework can be applied to the full range of prediction scenarios since these methods learn spatiotemporal representations autoregressively (and thus numbers are reported in both tables). We added a clarification to the baseline section (cf. last paragraph, Section 4.1, revised version).
> >
> > *In Table 3, VRNN does not appear in classification results. Even though VRNN is a sequential generative model, it’s hidden state can also be used for classification.*
> >
> > Can you maybe elaborate more on how you’d use VRNN for classification tasks or point out some work where this is done? Do you mean an auxiliary classifier that operates on the hidden state when processing supervised multi-agent segments? Or do you mean the replacement of the reconstruction loss by some classification loss (in which case the model could be only trained on the labeled share of the data). In both cases, the performance would most likely be significantly worse than the tested methods in Table 3, since a vanilla VRNN version assumes independence across the social dimension.
> >
> > *The work addresses the limitations of the work in terms of application in Section 4 but does not discusses potential negative social impact of the work. In the reviewer's opinion, discussing potential negative social impact may not be necessary for this particular paper as it is primarily a methodology work and the social impacts can depend on specific implication. However, it would definitely be welcome if the author can discuss the paper's limitations from broader social perspectives, in the rebuttal.*
> >
> > We have a clear focus on team sports. Though dual use of the proposed technique is certainly possible (e.g., military domains), it may be a bit far fetched as we have no expertise in these domains and would only be able to deliver uneducated guesses about whether this is realistic or not. Generally, the datasets and tasks at-hand are privacy-sensitive as the targets are human individuals. While the soccer and basketball players are probably identifiable in the data, they/their contract/their clubs agreed in recording the data.
> >
> > We will move this discussion of the social impacts from the checklist to the conclusion upon acceptance.

---

> > > ### Comment · Reviewer_iUsD · 2022-08-08
> > > **Response after rebuttal**
> > >
> > > Thanks for the detailed response. I'm satisfied with the author's answer to my two questions and appreciate the efforts to summarize the work's contributions. I will keep my positive rating for this paper.

---

### Official Review · Reviewer_KKnc · 2022-07-11

**Rating:** 6
**Confidence:** 4
**Soundness:** 3 good
**Presentation:** 2 fair
**Contribution:** 2 fair

**Summary:**

This work presents a semi-supervised approach for modeling multi-agent trajectories in the conditional variational autoencoder (CVAE) framework. The main contribution is the formalization of the combined losses of (weakly) labeled and unlabeled data points in spatiotemporal multi-agent settings. Experiments are performed on the NBA dataset, Stanford Drone Dataset and an Elite soccer dataset not publicly available (as far as I can tell). The proposed approaches (S-MAT and U-MAT) show strong performance compared to a variety of baselines.

**Questions:**

- Why is the citation on line 114 anonymized? There are many prior works that have stated and showed that human movement is inherently multi-modal. This is why most work in trajectory prediction employs latent variable models and evaluations are typically done minimum over K predictions, where K is the number of modes.
- The sentence on line 143 needs to be proofread:
“… labeled or unlabeled observations, however, … obtained to also allowing …” → “… labeled or unlabeled observations. However, … obtained to also allow …”

- Line 165: “Arbitrary and possible random order” → “arbitrary order”?
- Can the authors clarify what they mean by: “order agnostic methods rely on the independence of agent trajectories”? Or possibly provide a reference that qualifies this statement. There are many prior approaches in multi-agent motion prediction that are order agnostic and do not rely on the independence of agent trajectories (for example, [2] [3] [4]).
- What is meant by “(un)supervised”? It is present throughout the text and it is unclear to me.
- Line 243: “Following related work” what’s the citation?
- Out of curiosity, is the elite soccer dataset a contribution that will be released as part of this work?
- Figure 2 seems to be based on the NBA dataset. However, it is referenced in the elite soccer experiments on line 314. Can the author please clarify this point?

References:

[2] Ngiam, Jiquan, Benjamin Caine, Vijay Vasudevan, Zhengdong Zhang, Hao-Tien Lewis Chiang, Jeffrey Ling, Rebecca Roelofs et al. "Scene transformer: A unified multi-task model for behavior prediction and planning." arXiv e-prints (2021): arXiv-2106.

[3] Casas, Sergio, Cole Gulino, Simon Suo, Katie Luo, Renjie Liao, and Raquel Urtasun. "Implicit latent variable model for scene-consistent motion forecasting." In European Conference on Computer Vision, pp. 624-641. Springer, Cham, 2020.

[4] Girgis, Roger, Florian Golemo, Felipe Codevilla, Martin Weiss, Jim Aldon D'Souza, Samira Ebrahimi Kahou, Felix Heide, and Christopher Pal. "Latent Variable Sequential Set Transformers for Joint Multi-Agent Motion Prediction." In International Conference on Learning Representations. 2021.



**Limitations:**

Limitations are addressed throughout the text. Societal impact is somewhat addressed in the Checklist; I believe this can be moved to the conclusion.

**Strengths And Weaknesses:**

Strengths

- I found the Related work section to be well-written and provided good coverage of the related fields.
- As far as I know, the formalization of semi-supervision in the context of CVAE for multi-agent trajectory prediction is novel.

Weaknesses
- The title of the paper is inaccurate. There is a plethora of latent variable models proposed for multi-agent trajectories. And the main part that sets this work apart from prior work is the use of semi-supervision in the CVAE framework.
- I found it difficult to follow section 3, starting from 3.3 onwards. For instance:
    - y_tilde appears on line 125 without any reference in the equations above.
    - Figure 1 is referenced once at the end of section 3.5, and its caption is very short and not descriptive.
    - I believe the second part of Equation 2 seems to be missing z_{<t} on the left-hand side.
    - I found the last paragraph of section 3.4 (line 158) unclear. I think this explanation can be improved.
- Having read section 4, I still found it unclear what the labels y correspond to in the supervised and unsupervised cases. For instance, what are the 3 agent types in the U-MAT? Also, my understanding is that S-MAT uses y_t as the area the agent should be in a timestep t. Is this correct? If so, this seems in line with goal-conditioned motion forecasting which has been proposed in prior work (e.g., [1]).
- I’m not sure I follow the argument starting at line 247. If the work in question (e.g., DAG-Net) is using goals that are predicted by the model, why does that render it inappropriate for comparison with unsupervised methods? If everything is predicted, i.e., no ground-truth goal information is used, then it seems fair to me. I would like to also confirm with the authors that S-MAT does not use ground-truth $y_{\leq T}$ and only uses the observed ground-truth labels.
- There are newer methods that evaluated on the NBA sportVU dataset using a different a different configuration (5 observation timesteps, 10 prediction timesteps). I refer the authors to Table 3 of the EvolveGraph paper ([31] in manuscript). The reason I mention this is because EvolveGraph is one of the stronger methods in the field and it would help to compare the proposed approach to this method.
- The experiments in Section 4.4 on the elite soccer dataset seem to present a marginal improvement over the VRNN baseline.

References:

[1] Zhao, Hang, Jiyang Gao, Tian Lan, Chen Sun, Benjamin Sapp, Balakrishnan Varadarajan, Yue Shen et al. "Tnt: Target-driven trajectory prediction." arXiv preprint arXiv:2008.08294 (2020).

---

> ### Author Response · Authors · 2022-08-02
> **Response to Reviewer KKnc**
>
> We thank the reviewer for the detailed review as well as the suggestions for improvement. We hope our response alleviates your concerns.
>
> *The title of the paper is inaccurate. There is a plethora of latent variable models proposed for multi-agent trajectories. And the main part that sets this work apart from prior work is the use of semi-supervision in the CVAE framework.*
>
> Previous VAE-based modeling approaches in the spatiotemporal regime employ either unsupervised generative settings, in which the latent space is described fully by discrete [13, 49, 31, 28, 16, 14] or continuous [28, 52, 59, 16, 14, 33] variables, or alternatively are formalized in a supervised framework where weakly preserved ``macro-intents" act as explicitly defined discrete information while the continuous latent subspace is learned from the data [64, 65, 40]. At a high level, our work subsumes these approaches into a unified framework via using an disentangled latent space that allows (discrete) supervision signals (or more generally, behavioral indicators) in arbitrary quantities to govern the generative process. The proposed method thus targets a more general set of problems so we opted for this title. However, we also agree that the title might mislead from the papers' main formulation and will ask the program chair to change the title of the paper to "Semi-Supervised Generative Models for Multi-Agent Trajectories" after acceptance.
>
> *I found it difficult to follow section 3, starting from 3.3 onwards. For instance:*
>
> - *y_tilde appears on line 125 without any reference in the equations above.*
>
> For the inference/encoding part, we need to differentiate between observed and latent discrete factors, which is why we introduce an indicator for unobserved labels at this point. We have updated the manuscript to introduce $\tilde{y}$ properly (and made clear that it is not referring to the equations above).
>
> - *Figure 1 is referenced once at the end of section 3.5, and its caption is very short and not descriptive.*
>
> Thanks. We have updated the illustration to clarify the relationships to the derived theorems in Section 3.3. and added a more detailed caption and reference (cf. revised manuscript or anonymous link: [https://github.com/hghfghfhhdc/neurips_rebuttal/blob/main/updated_architecture.png](https://github.com/hghfghfhhdc/neurips_rebuttal/blob/main/updated_architecture.png)).
>
> - *I believe the second part of Equation 2 seems to be missing z_{<t} on the left-hand side.*
>
> Do you mean that $z_{<t}$ should be included in the condition of the joint distribution? Note that we only consider z realizations from the past when factorizing over time.
>
> - *I found the last paragraph of section 3.4 (line 158) unclear. I think this explanation can be improved.*
>
> $L(x,y)$ can be interpreted as a reconstruction loss, since $p_\theta(x_t|.)$ is the largest factor in L (the KL divergence and $p(y|.)$ terms tend to exhibit smaller loss values, relatively speaking). We take the expectation of L wrt. $q_\phi(\tilde{y}|.)$. Thus, the model is encouraged to assign the highest probability mass for the latent label value that yields the lowest L value/reconstruction loss. This is a desirable property that reflects our assumption of data generation. We made clarifying edits in the manuscript (now line 165-169).
>
>
>
> - *I’m not sure I follow the argument starting at line 247. If the work in question (e.g., DAG-Net) is using goals that are predicted by the model, why does that render it inappropriate for comparison with unsupervised methods? If everything is predicted, i.e., no ground-truth goal information is used, then it seems fair to me. I would like to also confirm with the authors that S-MAT does not use ground-truth y≤T  and only uses the observed ground-truth labels.*
>
> During the observation period (10 timesteps), [40, 64] do not use predicted goals, but rather rely on ground-truth information inferred in advance to model training and predict everything in the remaining 40 timesteps (including goals). We use exactly the same setup in our S-MAT experiments, which seems valid at first sight. However, these labels encode future information. For example, y_10 (the last observed label) includes positional information from some future state. We edited line 248 to clarify this.

---

> > ### Author Response · Authors · 2022-08-02
> > **Response to Reviewer KKnc (cont.)**
> >
> > - *Having read section 4, I still found it unclear what the labels y correspond to in the supervised and unsupervised cases. For instance, what are the 3 agent types in the U-MAT?*
> >
> > First, we observe (e.g., line 246-248, 32-40) that existing SOTA trajectory prediction approaches (Section 4.1., second paragraph) use heuristically generated labels for trajectory prediction that encode agents' intents or goals over a discretized position space (visualized in Figure 2 right; computation explained in line 220-224). We then observe that our formalization allows to naturally integrate such long-term goals into the overall scheme via treating them as semantic concepts y_t (Section 4.2., line 239-245). We added citations for clarity (line 243) and also a reference pointing to Section 4.1. Since these labels are generated heuristically based on trajecotry data prior to model training, this instantiation of our framework is fully-supervised and is referred to as S-MAT (first paragraph, Section 4.2.).
> >
> > However, we found (although done in [40, 64]) that benchmarking against unsupervised baselines is inappropriate for the reasons described in lines 246-254. Since most SOTA approaches are unsupervised generative models, we additionally propose a fully-unsupervised instantiation of our framework (U-MAT). In the fully unsupervised case, the model learns to structure the discrete subspace by itself without any annotation. From the predicted (latent) categorical distribution $q_\phi(\tilde{y}|.)$, we sample a label value and exploit separate motion predictors (different decoder parameterizations) $p_\theta(x_t|.)$ dependent on the realized y-value. Intuitively, this encourages the model to learn categories  describing fundamentally different movement patterns, which can be interpreted as dynamic “agent roles”. Thus, the concept of agent roles here is merely an intuitive explanation and realized via an inductive bias that increases the “scope” of latent information encoded without utilizing any supervision. To the best of our knowledge, parameterizing the generation module based on inferred agent categories is novel and could provide valuable insights for practitioners.
> >
> > We added more detailed model descriptions to Appendix C. The key point of Section 4.2. is to validate our proposed architecture and we refer to [40, 64] for more details on how to produce the weak labels used for training S-MAT.
> >
> > - *Also, my understanding is that S-MAT uses y_t as the area the agent should be in a timestep t. Is this correct? If so, this seems in line with goal-conditioned motion forecasting which has been proposed in prior work (e.g., [1]).*
> > [1] Zhao, Hang, Jiyang Gao, Tian Lan, Chen Sun, Benjamin Sapp, Balakrishnan Varadarajan, Yue Shen et al. "Tnt: Target-driven trajectory prediction." arXiv preprint arXiv:2008.08294 (2020).
> >
> > There is a body of research that uses long-term intents as a means to improve trajectory prediction (in addition to Hang et al. [1], e.g., [37, 38] for vehicles and pedestrians and [40, 64] in the sports domain). We propose a general framework in which any type of discrete behavioral indicators, including such long-term goals, can be naturally integrated. Thus, indeed, S-MAT can be thought of some version of goal-condition motion forecasting.
> >
> > A key difference, however, is how these intents are calculated. While Hang et al. [1] and [37, 38] compute the agent goals based on fixed time horizons, our setting defines them as stationary points in the (discretized) position (as proposed in [40] and described previously). In our opinion, the use of fixed time windows such as endpoints is too rigid for highly interactive and highly nonlinear sports datasets, since game plays consist of many intermediate and dynamic stationary points. Empirically, an indication of this can be found in [36], where [40] performs significantly worse on basketball data than its non goal-conditioned competitors (while achieving SOTA performance for non-interactive pedestrian datasets).
> >
> > - *There are newer methods that evaluated on the NBA sportVU dataset using a different configuration (5 observation timesteps, 10 prediction timesteps). I refer the authors to Table 3 of the EvolveGraph paper ([31] in manuscript). The reason I mention this is because EvolveGraph is one of the stronger methods in the field and it would help to compare the proposed approach to this method.*
> >
> > Although an additional comparison to EvolveGraph [31] would be indeed interesting, it is infeasible for the following reasons:
> > - Firstly, there is no publically available source code.
> > - Comparing against their reported numbers is likewise impossible as there is no reference to the data “collected by NBA with the SportVU tracking system” (cf. 8.1.2 in [31]) and their experimental setup remains ambigous (and is tailored by the authors).
> > Furthermore, we note that GRIN [33] (one of our used baselines) is a more recent method in this field.

---

> > > ### Author Response · Authors · 2022-08-02
> > > **Response to Reviewer KKnc (cont.)**
> > >
> > > - *The experiments in Section 4.4 on the elite soccer dataset seem to present a marginal improvement over the VRNN baseline.*
> > >
> > > Although the results are generally more similar compared to the basketball experiments, we would like to point out the following considerations:
> > > - For soccer dynamics, due to the bigger field and higher sampling rate (in our case, 25 Hz), players tend to run more linearly compared to basketball. Hence, we hypothesize that models with implicit independence assumptions across agents (and consequently no social interaction module), such as the VRNN baseline, tend to produce stronger results compared to basketball experiments since the past trajectory is frequently more informative for predicting its future than agent-agent dependencies (relatively speaking). This is also reflected in the comparatively small improvements when benchmarking interactive models against their (socially) independent peers (cf. VRNN vs. VRNN + GNN and MAT-Diag vs. MAT in Table 3).
> > > - As stated in line 295-297, the VRNN constitutes a strong competitor for the task at-hand since it effectively corresponds to a diagonal version of GVRNN [59] (referred to as GVRNN-Diag in [59]), which has been shown to produce competitive results on soccer data [59]. In fact, although the sampling rate is lower, the realized improvements of the best model in [59] are inferior to those in our work (3.11 vs. 3.07 for avg L_2, see Table 3 in [59]). We added GVRNN results to our manuscript in Table 3 (left side) which empircally validate this (VRNN: 2.91 vs. GVRNN: 2.89 vs. MAT (Ours): 2.81 for avg L_2).
> > > - As already indicated, considering the fairly high sampling rate and the resulting inherent lower prediction error, the observed improvements are actually reasonable. We hypothesize that decreasing the frequency of the data and extracting multi-agent segments that merely consist of highly interactive plays significantly yields an increase in performance gap.
> > >
> > > We added more details to the corresponding Section in the revised version.
> > >
> > > *The sentence on line 143 needs to be proofread: “… labeled or unlabeled observations, however, … obtained to also allowing …” → “… labeled or unlabeled observations. However, … obtained to also allow …”*
> > > *Line 165: “Arbitrary and possible random order” → “arbitrary order”?*
> > > Thanks, we corrected the sentences.
> > >
> > > *Can the authors clarify what they mean by: “order agnostic methods rely on the independence of agent trajectories”? Or possibly provide a reference that qualifies this statement. There are many prior approaches in multi-agent motion prediction that are order agnostic and do not rely on the independence of agent trajectories (for example, [2] [3] [4]).*
> > >
> > > [2] Ngiam, Jiquan, Benjamin Caine, Vijay Vasudevan, Zhengdong Zhang, Hao-Tien Lewis Chiang, Jeffrey Ling, Rebecca Roelofs et al. "Scene transformer: A unified multi-task model for behavior prediction and planning." arXiv e-prints (2021): arXiv-2106.
> > >
> > > [3] Casas, Sergio, Cole Gulino, Simon Suo, Katie Luo, Renjie Liao, and Raquel Urtasun. "Implicit latent variable model for scene-consistent motion forecasting." In European Conference on Computer Vision, pp. 624-641. Springer, Cham, 2020.
> > >
> > > [4] Girgis, Roger, Florian Golemo, Felipe Codevilla, Martin Weiss, Jim Aldon D'Souza, Samira Ebrahimi Kahou, Felix Heide, and Christopher Pal. "Latent Variable Sequential Set Transformers for Joint Multi-Agent Motion Prediction." In International Conference on Learning Representations. 2021.
> > >
> > > This statement refers to the adjustments required to process trajectory data involving multiple agents at this point in the manuscript. It implies that if we were to use the previously derived objective functions directly (which at this point only capture temporal dependencies, cf. equations (5) and (6)), independence over the agent dimension is required to achieve permutation-invariance which in turn is a prerequisite for processing sequences of unordered collections like multi-agent trajectories. This independence assumption is trivially inappropriate for interactive systems, so work in this area (besides the above works [2,3,4], the cited trajectory papers in our related work section) proposes sensitive solutions (usually in the form of some graph encoding strategy) to capture agent-agent interactions. We agree that the statement is misleading in the sense that we are the first to model agent dependencies while maintaining a permuation-invariant model. However, it merely acts as a "step-by-step” explanation for deriving the final architecture and we have edited Section 3.5. for clarity. A discussion wrt. this topic can be found in e.g. [59].

---

> > > > ### Author Response · Authors · 2022-08-02
> > > > **Response to Reviewer KKnc (cont.)**
> > > >
> > > > *Why is the citation on line 114 anonymized? There are many prior works that have stated and showed that human movement is inherently multi-modal. This is why most work in trajectory prediction employs latent variable models and evaluations are typically done minimum over K predictions, where K is the number of modes.*
> > > >
> > > > Indeed, almost all work on trajectory prediction mentions the non-determenistic and multimodal nature of interactive multi-agent systems and thus argues in favor of deep latent variable models and min metrics over multiple modes. We deleted the reference.  (We are not sure if this was a question, but we note that we also reported the minimum error over $K=20$ modes due to this exact reason).
> > > >
> > > > *What is meant by “(un)supervised”? It is present throughout the text and it is unclear to me.*
> > > >
> > > > It means (fully) supervised and unsupervised.
> > > >
> > > > *Line 243: “Following related work” what’s the citation?*
> > > >
> > > > The papers are [40,64], which use the weakly preserved labels described in Section 4.1. We added the citations.
> > > >
> > > > *Out of curiosity, is the elite soccer dataset a contribution that will be released as part of this work?*
> > > >
> > > > Unfortunately, we cannot release the data but will add a pointer to the company that owns the rights.
> > > >
> > > > *Figure 2 seems to be based on the NBA dataset. However, it is referenced in the elite soccer experiments on line 314. Can the author please clarify this point?*
> > > >
> > > > Thanks! We corrected line 314 (we confused “left” and “right” in the text) and added caption descriptions.
> > > >
> > > > *Limitations are addressed throughout the text. Societal impact is somewhat addressed in the Checklist; I believe this can be moved to the conclusion.*
> > > >
> > > > As suggested, we will move this discussion of the social impacts from the checklist to the conclusion upon acceptance:
> > > >
> > > > We have a clear focus on team sports. Though dual use of the proposed technique is certainly possible (e.g., military domains), it may be a bit far fetched as we have no expertise in these domains and would only be able to deliver uneducated guesses about whether this is realistic or not. Generally, the data and tasks at-hand are privacy-sensitive as the targets are human individuals. While the soccer and basketball players are probably identifiable in the data, they/their contract/their clubs agreed in recording the data.

---

> > > > > ### Comment · Reviewer_KKnc · 2022-08-08
> > > > > **Response to Rebuttal**
> > > > >
> > > > > I thank the authors for their thorough response to my review. I am quite satisfied with the improvements made to the manuscript and will be increasing my score to weak accept.
> > > > >
> > > > > I invite the authors to perform another round of proofreading before camera-ready should the manuscript be accepted. For example:
> > > > > - $\lambda_2$ reference on line 162.
> > > > > - "costumize" on line 181.
> > > > > - "descired" on line 182.
> > > > >
> > > > > I would also ask the authors to improve Figure 1 in the camera ready version of this work. Please ensure that parts of the figure are not overlapping (prior_i versus encoder_i). I also suggest that fig. 1 (right) be split into two separate (sub-)figures, one for encoding and one for generative. I find the number of arrows difficult to follow.

---

### Official Review · Reviewer_DZJp · 2022-07-15

**Rating:** 5
**Confidence:** 4
**Soundness:** 3 good
**Presentation:** 3 good
**Contribution:** 3 good

**Summary:**

This paper presents a latent variable model for modeling multi-agent trajectories. The authors present a generative framework for modeling and annotating trajectories of multiple agents over time and space. Their approach leverages semi-supervised learning for variational autoencoders where equivariance and interaction of agents are appropriately represented by graph networks. The proposed approach is generally applicable to a wide range of problems, including generation, classification, and pattern detection. Empirically, the proposed approach outperforms all competitors on various interactive spatiotemporal problems.


**Questions:**

The authors are suggested to address the comments and concerns in the "weaknesses" and "limitations" sections in their rebuttal.

**Limitations:**

There seems no discussion on the limitations and potential negative societal impact of this work. It would be better to provide some discussions on these aspects in the rebuttal and revised version.

**Strengths And Weaknesses:**

Strengths:
1. This paper is generally well written and easy to follow. The structure and presentation of this paper are clear.
2. Generative modeling of multi-agent trajectories is an interesting direction to investigate.

Weaknesses:
1. The technical novelty of this paper needs to be clarified more clearly. The proposed framework in Figure 1 includes two encoders to obtain both discrete and continuous latent variables, which was already proposed in [27]. It would be better to explicitly summarize the major novelty of this work and how it differs from existing methods.
2. For trajectory generation tasks, how do you obtain the labels for latent variable $y$? Is it labeled by humans or automatically based on heuristics? The generation task in semi-supervised settings should be introduced more clearly.
3. Since manual human annotations of events are used in this work, it would be better to also conduct an ablation study on using different portions of the annotated data for training with the purpose of validating how well the proposed method can leverage the small set of annotated data.
4. In the references section, it would be better to replace arXiv versions with the actual publication conferences/journals. Another suggested related paper for relational inference is listed below:
[A] RAIN: Reinforced Hybrid Attention Inference Network for Motion Forecasting, ICCV 2021.

---

> ### Author Response · Authors · 2022-08-02
> **Response to Reviewer DZJp**
>
> We would like to thank you for your hard work and constructive feedback. We hope that the following response aids in mitigating your concerns.
>
> *The technical novelty of this paper needs to be clarified more clearly. The proposed framework in Figure 1 includes two encoders to obtain both discrete and continuous latent variables, which was already proposed in [27]. It would be better to explicitly summarize the major novelty of this work and how it differs from existing methods.*
>
> Having a disentangled latent space with a discrete and continous subpart is the standard setup for semi-supervised generative models. However, all approaches in the literature are introduced only for static domains (mostly images). Our contribution could be seen as a spatiotemporal generalization of [27]. However, this generalization is not straight forward but instead requires modeling two additional dimensions (time and space; see first paragraph of Section 2). To clarify the relationship to [27] and to highlight our novelities, we have added the right part to Figure 1 and updated our architecture visualization (left part, Figure 1), extended its caption, made general clarifying edits on the novelities/differences throughout the paper (Abstract; Sections 1, 2 and 3), and altered section titles (cf. Section 3).
>
> To summarize, our work is distinct from others in the following ways:
>
> - We formalize a probabilistic generative model in spatiotemporal regimes by a semi-supervised approach. This allows us to
>     - Model simultaneously disentangled discrete (defined by the user of the system, e.g., expert knowledge) and continuous effects. Previous VAE-based spatiotemporal models employ either unsupervised generative settings, in which the latent space is described fully by discrete [13, 49, 31, 28, 16, 14] or continuous [28, 52, 59, 16, 14, 33] variables, or, are alternatively formalized in a supervised framework where weakly preserved ``macro-intents" act as explicitly defined discrete information while the continuous latent subspace is learned from data [64, 65, 40]. At a high level, our contribution subsumes these approaches into a unified framework via a disentangled latent space that allows (discrete) supervision signals (or more generally, behavioral indicators) in arbitrary quantity and annotation type to govern the generative process. Intuitively, this increases the "scope" of latent information captured by the method and improves upon existing SOTA methods in trajectory forecasting settings.
>     - Lift applicability to highly relevant prediction problems beyond representational or generative modeling. For example, intrinsic to our approach is an encoding module that argues over the label space and can thus be used in real-world settings with expensive manual annotations. We validated the advantages empircally. Existing generative models for spatiotemporal pattern detection focus almost exlusively on mulit-agent trajectory prediction.
> - Though the methodological tools (RNN+GNN) have been used before, the concrete architectures that emerge based on the derived theorems and practical considerations (cf. e.g., Section 3.5.) are novel. The architectual design choices are validated empirically via outperforming SOTA baselines (Section 4, Appendix D) and ablation studies (Appendix E).
>
> **Note:** The updated Figure can be also accessed through the following anonymous link: [https://github.com/hghfghfhhdc/neurips_rebuttal/blob/main/updated_architecture.png](https://github.com/hghfghfhhdc/neurips_rebuttal/blob/main/updated_architecture.png)
>
>
> *In the references section, it would be better to replace arXiv versions with the actual publication conferences/journals. Another suggested related paper for relational inference is listed below: [A] RAIN: Reinforced Hybrid Attention Inference Network for Motion Forecasting, ICCV 2021.*
>
> Thank you for pointing us to this work. We have included [A] in the related work section in the revised version and will update all references with their actual publication upon acceptance of this paper.
>
> *There seems no discussion on the limitations and potential negative societal impact of this work. It would be better to provide some discussions on these aspects in the rebuttal and revised version.*
>
> We have a clear focus on team sports. Though dual use of the proposed technique is certainly possible (e.g., military domains), it may be a bit far fetched as we have no expertise in these domains and would only be able to deliver uneducated guesses about whether this is realistic or not. Generally, the data and tasks at-hand are privacy-sensitive as the targets are human individuals. While the soccer and basketball players are probably identifiable in the data, they/their contract/their clubs agreed in recording the data.
>
> We will move this discussion of the social impacts from the checklist to the conclusion upon acceptance.

---

> > ### Author Response · Authors · 2022-08-02
> > **Response to Reviewer DZJp (cont.)**
> >
> > *For trajectory generation tasks, how do you obtain the labels for latent variable y? Is it labeled by humans or automatically based on heuristics? The generation task in semi-supervised settings should be introduced more clearly.*
> >
> > As stated in 3.2., the discrete subspace may be best thought of disciminative causes of variation and may contain anything that aids in generating multi-agent rollouts, including behavioral indicators, long-term goals (e.g., heuristically generated), or current tasks (e.g., manually annotated). To validate this general formulation and our proposed architecture, we therefore vary the concrete definition of the labels across our experiments in the respective sections.
> >
> > **Section 4.2.: Exploring Boundary Cases for Trajectory Prediction**
> >
> > First, we observe (e.g., line 246-248, 32-40) that existing SOTA trajectory prediction approaches (Section 4.1., second paragraph) use heuristically generated labels for trajectory prediction that encode agents' intents or goals over a discretized position space (visualized in Figure 2 right; computation explained in line 220-224). We then observe that our formalization allows to naturally integrate such long-term goals into the overall scheme via treating them as semantic concepts y_t (Section 4.2., line 239-245). We added citations for clarity (line 243) and also a reference pointing to Section 4.1. Since these labels are generated heuristically based on trajecotry data prior to model training, this instantiation of our framework is fully-supervised and is referred to as S-MAT (first paragraph, Section 4.2.).
> >
> > However, we found (although done in [40, 64]) that benchmarking against unsupervised baselines is inappropriate for the reasons described in lines 246-254. Since most SOTA approaches are unsupervised generative models, we additionally propose a fully-unsupervised instantiation of our framework (U-MAT). In the fully unsupervised case, the model learns to structure the discrete subspace by itself without any annotation. From the predicted (latent) categorical distribution $q_\phi(\tilde{y}|.)$, we sample a label value and exploit separate motion predictors (different decoder parameterizations) $p_\theta(x_t|.)$ dependent on the realized y-value. Intuitively, this encourages the model to learn categories  describing fundamentally different movement patterns, which can be interpreted as dynamic “agent roles”. Thus, the concept of agent roles here is merely an intuitive explanation and realized via an inductive bias that increases the “scope” of latent information encoded without utilizing any supervision. To the best of our knowledge, parameterizing the generation module based on inferred agent categories is novel and could provide valuable insights for practitioners.
> >
> > We added more detailed model descriptions to Appendix C. The key point of Section 4.2. is to validate our proposed architecture and we refer to [40, 64] for more details on how to produce the weak labels used for training S-MAT.
> >
> > **Section 4.4.: Combining Generation and Classification in Semi-Supervised Settings**
> >
> > This set of experiments mainly serves the purpose to quantitatively validate the benefits of incorporating expensive human annotations as a generation guide/for structuring the discrete posterior (MAT in Table 3) compared to fully-supervised approaches based on heuristic surrogates (introduced before, S-MAT in Table 3) or unsupervised generative models (VRNN, VRNN+GNN, GVRNN [59]). This approach comes with little additional costs since the method operates semi-supervised and therefore requires only a small subset of multi-agent segments to be annotated (here, we use 20% annotated data). We agree that MAT would benefit from a clearer introduction and description and we added an clarifying paragraph to Section 4.4..

---

### Author Response · Authors · 2022-08-02
**General Response to Reviewers**

We would like to thank all reviewers for their time and insightful feedback. In particular, we would like to acknowledge the positive comments on: novelty of the work (R2, R4), strong methodological motivation and literature background (R2, R3), the extensive experiments (R3, R4), relevance of the application addressed (R3, R4), contribution to the field of generative models (R1), and presentation (R1, R3). We have addressed the concerns and suggestions raised by the reviewers and revised the manuscript accordingly.

Thank you all for your time and consideration. Unless a citation is provided within individual responses, references refer to the reference numbers in the revised version of the manuscript. Please find invidual replies below.

---

### Meta-Review · Area_Chair_SG2W · 2022-08-25

**Recommendation:** Accept
**Confidence:** Certain

**Metareview:**

This paper studies an interesting problem, and overall the reviewers agreed the exposition and validation are sufficient. We encourage the authors to consider the issues raised by the reviewers and further improve the work in the final version.

**Award:**

No

---

### Decision · Program_Chairs · 2022-09-14

Accept